# Personality Traits and Family Environment: Antecedents of Child Aggression

**DOI:** 10.3390/brainsci12111586

**Published:** 2022-11-20

**Authors:** Ayoob Lone, Azzam Othman Albotuaiba

**Affiliations:** Department of Clinical Neurosciences, College of Medicine, King Faisal University, Al-hasa 31982, Saudi Arabia

**Keywords:** big five personality, family environment, aggression, Saudi children

## Abstract

Previous research reported significant relationship between Big Five personality traits and aggression in both adolescent’s and adult population. However, it is unclear about whether similar connection exist in early age. This study investigated how personality traits and family environment influence the development of aggression in school aged Saudi children. A sample of 315 school going children were recruited voluntarily to complete a set of measures examining Big Five personality factors, family environment and aggression. Correlation analysis were employed to evaluate association between Big Five personality traits, family environment and aggression. The results showed a significant relationship of Big Five personality factors and family environment factors to aggression. The findings of the study revealed that amongst the Big Five personality traits conscientiousness, agreeableness, and neuroticism were strongest antecedents of childhood aggression. Family environment characterized by family cohesion, expressiveness and conflict were appeared to be significant antecedents of aggressive behavior in children. This study expands our knowledge about the intervention strategies of aggression from Big Five traits and family environment perspectives.

## 1. Introduction

Aggressive behavior is long-standing issue that affects substantial number of young people. The appearance of their manifestations has drawn the attention of both the researchers and institutions of education. The conceptual and scientific approaches ought to offer suggestions for preventative strategies by creating numerous initiatives intended at reducing aggressive behavior among children. Aggressive behavior defined as disruptive behavior that harms people physically and psychologically [1]. Aggression can be manifested in many forms including physical aggression, verbal aggression, anger, and hostility [2]. Children’s aggressive behavior can have a number of negative consequences. On the one hand, children who demonstrate overt aggression and violent behavior may experience unpleasant emotions, resulting in poor attention and academic achievement [3,4,5]. On the other hand, aggressive behavior is strongly linked to peer victimization and bullying [6]. Bullying victims could experience more internalizing issues and even suicide thoughts which can have serious repercussions [7].

Various factors have been identified to influence on the emergence of aggressive behavior in children, e.g., personality traits such as anger, revengefulness, suspiciousness, etc., [8]. And social factors such as family and school environment, friends, social media, movies and computer games [9]. Many characteristics related to these factors may also play a vital role in the development of aggressive behavior among children, e.g., the type of family where the child is raised, parental methods, socioeconomic status of the family including parental education, occupation and economic standing, the neighborhood where the family lives, etc.

Personality traits are characterized as behavioral and psychological tendencies that remain stable over time and across different contexts [10]. The main indicator of personality functioning is the Big Five personality factors which describes five broad personality traits: Neuroticism, Extraversion, Openness to Experience, Agreeableness, and Conscientiousness [10]. Previous research separately investigated the relationship between the Big Five personality traits and situational/dispositional aggression. In situation-induced method neuroticism was positively and significantly associated with aggression, whereas agreeableness was negatively linked with aggression [11]. Regarding dispositional aggression, Dam et al. reported that aggression was positively linked with neuroticism while as agreeableness and conscientiousness were negatively correlated with aggression [12]. Li et al. indicated that neuroticism trait was positively associated with aggressive behavior while as agreeableness was negatively related with aggression [13,14]. Openness to experiences trait of Big Five appears to be unrelated to aggression [15]. However, the link between extraversion and aggression is complicated. Sharpe and Desai discovered a negative correlation between aggression and extraversion [16], while as Gallo and Smith reported that extraversion was positively and significantly correlated with aggression [17]. Finally, the association between the Big Five personality traits and dispositional aggression was slightly complicated, although the relationship between neuroticism, agreeableness, and aggression remained reasonably consistent. Therefore, the present study provides the new insights in associations between Big Five personality factors and aggression.

Family environment has been consistently associated with social and psychological problems in teenagers [18,19,20,21,22,23]. Previous literature demonstrated that bad family environment with more family conflict, poor communication, and low parental support has a negative impact on child’s overall development [24,25,26]. Studies on children in the middle grades found a direct link between family conflict and children’s aggressive behavior [27,28]. Tanaka et al.’s study found that family conflict was associated with higher proactive aggression with high levels of anxiety in children sample [29]. Another study suggested that poor child-parent relationship was significantly linked with child aggression and this negative relationship may develop a sense of insecurity and disappointment in later life [30]. Parents’ destructive conflict tactics and negative emotionality were linked to a higher risk of aggression in children [25]. It has been discovered that the bigger the number of risk variables to which an individual is exposed, the greater the likelihood that the individual may engage in violent behavior. It is crucial for prevention to recognize the factors that contribute to child aggression at the right stage of development [31]. Based on these insights, the aim of the present study was designed to examine the relationship of personality traits and characteristics of the family environment to aggression. We hypothesized that Big Five traits of personality and family environment will significantly predict aggression among school going children.

## 2. Materials and Methods

### 2.1. Participants

The participants of the present study were recruited from AlHasa Governorate of Saudi Arabia through convenience sampling. The sample for this study was calculated by the Raosoft online sample size calculator. The response distribution was assumed to be 50% with a confidence interval of 95% and a 5% margin of error. A total of 340 volunteer school going children were recruited to participate in this research and 315 children (Male = 147; Females = 168) aged between 4 and 12 years (*M* = 8.16; *SD* = 2.87) finally completed the survey. The other 25 study participants were reluctant to answer all items of the questionnaire were excluded.

### 2.2. Measurements

Personality factors were assessed by using Big Five Inventory-10 (BFI-10)) [32]. This scale comprises of 10 items covering five dimensions of personality: Extraversion, Agreeableness, Conscientiousness, Neuroticism, and Openness to Experience. Each subscale includes two items rated on 5-point Likert scale ranging from 1 (strongly disagree) to 5 (strongly agree). The scale is short and approximately takes less than 5 min to be filled. In this study, the reliability coefficient of five factors were 0.82 (Extraversion), 0.56 (Agreeableness), 0.65 (Conscientiousness), 0.84 (Neuroticism), and 0.66 (Openness to Experience).

Brief Family Relationship Scale [33] was used to measure the family environment. This 16-item scale comprises of three domains: Cohesion, Expressiveness, and Conflict. These subscales measure support, expression of opinions, and angry conflict within a family. Responses are obtained on a 3-point Likert scale ranging from 1 (Not at all) to 3 (A lot of me). Total score is produced by adding each subscale scores together. A high score on this measure indicates good family relationship. Fok, Allen, Henry, and Awakening Team obtained a reliability coefficient of 0.83, 0.80, 0.65, and 0.88 for cohesion, conflict, expressiveness, and full-scale BFRS, respectively [33]. The Cronbach’s alpha of this measure was found 0.78 (cohesion), 0.76 (expressiveness), 0.80 (conflict), and 0.86 (overall scale).

Aggression among children was measured by using Buss-Perry Aggression Questionnaire-Short Form (BPAQ-SF) [2]. The BPAQ-SF is a 12 items measuring aggression of respondents in four areas: physical aggression, verbal aggression, anger, and hostility [34]. There are three questions for each of the four factor. Responses are obtained on a 5-point Likert scale ranging from 1 (extremely uncharacteristic of me) to 5 (extremely characteristic of me). Total score is produced by adding each subscale scores together. A high score on this measure indicates more aggression. In the current study, the reliability coefficient (Cronbach’s alpha) of this scale was 0.82.

Finally, the present study used demographic questionnaire, including age, gender, grade, family structure, parental education, and socioeconomic status.

### 2.3. Procedure

After seeking required permission from ethics committee of scientific research, college of medicine, King Faisal University, Saudi Arabia., the participants were personally contacted. They were briefed about the purpose of research and questionnaire used in the study. After seeking consent from the children’s parent and children himself, a suitable time and date was fixed for data collection. Before administering the questionnaire, the purpose of the study was again explained to the participants and they were assured that their responses will be kept confidential and will be used only for research and academic purpose only. A good rapport was built with the participants for getting correct responses. After this, the questionnaires were provided to them and they were requested to fill-up the questionnaire as per the instructions given in the questionnaire. Research team helped these participants in understanding the items in the questionnaire. It took an average of 15 min for the participants to complete the questionnaire. After completion of the questionnaire participants returned the questionnaire and they were thanked for their participation and cooperation.

### 2.4. Statistical Analysis

After checking for quality and consistency of the data, then data was coded and put into EPI data version 3.1 and after that it was exported to SPSS (version, 25) for analyses. At first, frequency distribution and descriptive statistics with mean and standard deviation was computed. Second, Pearson’s correlation analysis was used to examine the relationship between personality factors, family environment and aggression. Finally, stepwise multiple regression analysis was applied to explore the role of personality factors and family environment in predicting aggression in the sample, and *p* ≤ 0.05 was considered as statistically significant.

## 3. Results

### 3.1. Initial Analysis

Participants of the present study (Table 1) were 315 children (147 males and 168 females) living in Hofuf city of Alhasa Region of Saudi Arabia. The age of these partici-pants ranged from 5 to 12 years (Mean = 8.45; SD = 2.67). About 86 (27.3 %) sample were studying in grade 6. Majority of the students belong to urban areas. Only 30 (9.50%) participants were residents of rural areas. The percentages of participants belonging to nuclear and joint families were 79.0 and 21.0 respectively.

### 3.2. Correlational Analysis

Pearson’s coefficient of correlation was used to examine the relationship between scores on Big Five personality factors, family environment and aggression. Coefficient of correlations (Table 2) of aggression with Big Five Factors revealed extraversion was negatively and significantly related to hostility (*r =* −0.12, *p* < 0.05). Agreeableness was negatively and significantly related to physical (*r =* −0.29, *p* < 0.01), verbal (*r =* −0.26, *p* < 0.01), anger (*r =* −0.19, *p* < 0.01) and hostility (*r =* −0.13, *p* < 0.05). Similarly, conscientiousness was negatively and significantly related to physical (*r =* −0.25, *p* < 0.01), verbal (*r =* −0.20, *p* < 0.01), anger (*r =* −0.22, *p* < 0.01) and hostility (*r =* −0.17, *p* < 0.05). However, neuroticism was found positively and significantly correlated with physical aggression (*r =* 0.21, *p* < 0.01), verbal aggression (*r =* 0.20, *p* < 0.01), anger (*r =* 0.31, *p* < 0.01) and hostility (*r =* 0.25, *p* < 0.01). This indicates that participants having neuroticism trait reported more aggressive in terms of physical, verbal, anger and hostility.

In regard to family environment of participants, cohesion was negatively and significantly related to physical aggression (*r =* −0.39, *p* < 0.01), verbal aggression (*r =* −0.31, *p* < 0.01), anger (*r =* −0.34, *p* < 0.01) and hostility (*r =* −0.46, *p* < 0.01). Similarly, expressiveness was negatively and significantly correlated with physical (*r =* −0.24, *p* < 0.01), verbal (*r =* −0.29, *p* < 0.01), anger (*r =* −0.19, *p* < 0.01) and hostility (*r =* −0.41, *p* < 0.01). This indicates that participants with good family cohesion and more expressive reported less physical aggression, verbal aggression, anger and hostility. However, conflict was found positively and significantly related to physical (*r =* 0.46, *p* < 0.01), verbal (*r =* 0.48, *p* < 0.01), anger (*r =* 0.45, *p* < 0.01) and hostility (*r =* 0.50, *p* < 0.01). This indicates that participants experience more family conflict reported more physical aggression, verbal aggression, anger and hostility.

### 3.3. Stepwise Multiple Regression

Stepwise multiple regression analysis was applied to examine the role of Big Five personality model and family environment in predicting children aggression. For this, separate analysis was run for each aspect of Big Five model and family environment. For the measure of Buss-Perry Aggression Questionnaire-Short Form (BPAQ-SF), results of stepwise multiple regression analysis presented in Table 3 clearly indicate that while predicting physical aggression from all factors of Big Five model, agreeableness (*β* = −0.29, *t* = −5.35, *p* < 0.01) could enter in the equation explaining 8% variance in physical aggression *R^2^* = 0.08, *F* (1, 313) = 28.69, *p* < 0.01. At step 2, conscientiousness entered in the equation which significantly predicted change in the scores on physical aggression (*β* = −0.21, *t =* −3.97, *p* < 0.01). Thus, this variable explained 4% variance in the criterion variable and the two variables jointly explained 12% variance in the scores on the dependent measure which was statistically significant *R*^2^ = 0.12, *F* (1, 313) = 22.94, *p* < 0.01. At step 3, the variable neuroticism significantly predicted change in the scores on physical aggression (*β =* 0.11, *t =* 2.03, *p* < 0.05) explaining 2% variance in the dependent measure and these three variables jointly explained 14% variance in the scores on physical aggression, *R*^2^ = 0.14, *F* (1, 313) = 16.81, *p* < 0.01. Negative relationship of agreeableness and conscientiousness with physical aggression indicate that children scoring high in agreeableness and conscientiousness showed less physical aggression whereas positive relationship of neuroticism was related to increase in physical aggression.

While predicting verbal aggression from scores on various factors of Big Five model, at step 1, agreeableness was found as significant predictor of verbal aggression (*β* = −0.25, *t* = −4.75, *p* < 0.01), which accounted for 6% variance in the scores on criterion variable *R^2^* = 0.06, *F* (1, 313) = 22.59, *p* < 0.01. At step 2, when conscientiousness was entered in the equation it significantly predicted change in the scores on verbal aggression (*β* = −0.17, *t* = 3.11, *p* < 0.05) explaining 3% variance in dependent measure. Both these variables jointly explained 9% variance in the score on verbal aggression which was significant *R^2^* = 0.09, *F* (1, 313) = 16.46, *p* < 0.01. At step 3, the variable of neuroticism entered in the equation which significantly predicted change in the scores on verbal aggression (*β =* 0.12, *t =* 2.16, *p* < 0.05), however, it could explain only 1% variance in the criterion variable. These three variables jointly explained 10% variance in the score on dependent measure *R^2^* = 0.10, *F* (1, 313) = 12.66, *p* < 0.01. Result revealed that agreeableness and conscientiousness were negatively related to verbal aggression, this mean that with high score in agreeableness and conscientiousness factor verbal aggression decreases significantly, whereas neuroticism was positively related to verbal aggression indicating participant with neuroticism trait showing more verbal aggression.

Neuroticism was found as significant predictor of anger (*β* = 0.31, *t =* 5.81, *p* < 0.01) which accounted for 9% variance in criterion variable *R^2^* = 0.09, *F* (1, 313) = 33.78, *p* < 0.01. When conscientiousness factor entered in the equation at step 2, it significantly predicted change in the scores on anger (*β* = −0.16, *t =* −2.98, *p* < 0.05) explaining 3% variance in the dependent variable. These two variables jointly explained 12% variance in the scores on anger which was significant *R^2^* = 0.12, *F* (3, 313) = 21.77, *p* < 0.01. At step 3, the variable of agreeableness entered in the equation which significantly predicted change in the scores on anger (*β* = −0.10, *t* = 1.97, *p* < 0.05), however, it could explain only 1% variance in the criterion variable. These three variables jointly explained 13% variance in the score on dependent measure *R^2^* = 0.13, *F* (1, 313) = 15.94, *p* < 0.01. These result revealed that conscientiousness and agreeableness were negatively related to anger, this means that with high score in conscientiousness and agreeableness factor anger among the participants decreases significantly, whereas neuroticism was positively related to anger indicating participant with neuroticism trait showing more anger.

While predicting hostility from scores on various factors of Big Five, at step 1, neuroticism was found as significant predictor of hostility (*β* = 0.25, *t =* 4.65, *p* < 0.01), which accounted for 6% variance in the scores on criterion variable *R*^2^ = 0.06, *F* (1, 313) = 21.64, *p* < 0.01. At step 2, when conscientiousness was entered in the equation it significantly predicted change in the scores on hostility (*β* = −0.12, *t =* −2.27, *p* < 0.05) explaining 2% variance in dependent measure. Both these variables jointly explained 8% variance in the score on hostility which was significant *R^2^* = 0.08, *F* (1, 313) = 13.55, *p* < 0.01. Result revealed that neuroticism was positively related to hostility, this indicates that children with high score in neuroticism factor hostility increases significantly, whereas conscientiousness was negatively related to hostility indicating participant with conscientiousness trait showing less hostility.

In regard to the family environment of participants results of stepwise multiple regression presented in Table 4 revealed that when predicting physical aggression from scores on brief family relationship scale, conflict was found significant (*β* = 0.46, *t* = 9.30, *p* < 0.01) predictor of physical aggression which explained 21% variance in dependent measure *R*^2^ = 0.21, *F* (1, 313) = 86.52, *p* < 0.01. When the variable of cohesion entered in the equation at step 2, it significantly predicted change in the scores on physical aggression (*β* = −0.23, *t =* −4.33, *p* < 0.01) explaining 5% variance in the dependent variable. These two variables jointly explained 26% variance in the scores on physical aggression which was significant *R^2^* = 0.26, *F* (1, 313) = 55.10, *p* < 0.01. Results revealed that conflict was positively related to physical aggression. This indicates that with increasing conflict physical aggression increasing significantly, However, negative relationship of cohesion with physical aggression indicted that participants with good family cohesion reported less physical aggression.

While predicting verbal aggression from scores on brief family relationship scale, at step 1, conflict was found significant (*β* = 0.48, *t* = 9.83, *p* < 0.01) predictor of verbal aggression which explained 23% variance in dependent measure *R^2^* = 0.23, *F* (1, 313) = 96.62, *p* < 0.01. At step 2, when the variable of expressiveness entered in the equation, it significantly predicted change in the scores on verbal aggression (*β* = −0.15, *t =* −3.09, *p* < 0.01) explaining 2% variance in the dependent variable. These two variables jointly explained 25% variance in the scores on verbal aggression which was significant *R^2^* = 25, *F* (1, 313) = 54.42, *p* < 0.01. Results showed that conflict was positively related to verbal aggression. This indicates that with increasing conflict verbal aggression increasing significantly, However, negative relationship of expressiveness with verbal aggression indicted that participants with high expression reported less verbal aggression.

Conflict was found as significant predictor of anger (*β* = 0.45, *t =* 8.91, *p* < 0.01) which accounted for 20% variance in criterion variable *R^2^* = 0.20, *F* (1, 313) = 79.54, *p* < 0.01. When cohesion factor entered in the equation at step 2, it significantly predicted change in the scores on anger (*β* = −0.17, *t* = −3.09, *p* < 0.01) explaining 2% variance in the dependent variable. These two variables jointly explained 22% variance in the scores on anger which was significant *R^2^* = 0.22, *F* (1, 313) = 45.66, *p* < 0.01. These result revealed that conflict were positively related to anger, this means that with high score in conflict factor anger among the participants increases significantly, whereas cohesion was negatively related to anger indicating participant with more cohesion showing less anger.

While predicting hostility from scores on brief family relationship scale, at step 1, conflict was found significant (*β* = 0.50, *t* = 10.25, *p* < 0.01) predictor of hostility which explained 25% variance in dependent measure *R^2^* = 0.25, *F* (1, 313) = 105.07, *p* < 0.01. At step 2, When the variable of expressiveness entered in the equation, it significantly predicted change in the scores on hostility (*β* = −0.29, *t* = −5.95, *p* < 0.01) explaining 7% variance in the dependent variable. These two variables jointly explained 32% variance in the scores on hostility which was significant *R^2^* = 0.32, *F* (1, 313) = 76.07, *p* < 0.01. At step 3, the variable of cohesion entered in the equation which significantly predicted change in the scores on hostility (*β* = −0.18, *t* = 2.96, *p* < 0.01), however, it could explain only 2% variance in the criterion variable. These three variables jointly explained 34% variance in the score on dependent measure *R^2^* = 0.34, *F* (1, 313) = 54.91, *p* < 0.01. Results showed that conflict was positively related to hostility. This indicates that with increasing conflict hostility increasing significantly, However, negative relationship of expressiveness and cohesion with hostility indicted that participants with high expression and good cohesion reported less hostility.

## 4. Discussion

The present study was carried out to examine the influence of personality characteristics and family environment factors on aggression. To the best our knowledge, this is a first study in Saudi Arabia which explores the personality factors using Big Five model and family environment in the development of aggression among school going children. The findings revealed a substantial link between Big Five personality characteristics and family environment components and aggression. The findings of the study revealed that conscientiousness, agreeableness, and neuroticism factors of personality were significantly related to aggression. These results are in line with previous investigations examined the relationship between personality factor and aggression in adolescent sample [15,35,36,37,38]. Among the different dimensions of Big Five model, agreeableness and conscientiousness were significantly and negatively correlated with aggression. Present findings are consistent with previous studies which showed relationship of agreeableness and aggression [39,40,41,42]. Agreeableness, which has been defined as cooperative and understanding [15] is a characteristic that is associated to the motivation to preserve favorable interpersonal connections [43]. Additionally, agreeableness indirectly reduces aggression by encouraging prosocial behavior and maintaining an optimistic and sympathetic view of human nature [42]. Costa and McCrae [44] confirmed that high degrees of agreeableness encourage prosocial behaviors like cooperativeness, kindness, and altruism, while low agreeableness encourage the inclination to feel less sympathy and empathy towards others [45]. Therefore, low agreeableness may enhance interpersonal conflict because it weakens social and relational control mechanisms leads the individual to quick and more easily act on their aggressive and violent impulses [11].

Consistent with numerous studies examining the relationship between conscientiousness trait and aggressive behavior [15,46,47], our findings indicated that conscientiousness linked negatively with aggression. Negative relationship of conscientiousness with aggression shows that children scoring high in conscientiousness trait exhibit less aggression. Moreover, negative relationship demonstrates that participants who are self-disciplined and have control over their impulses express less aggression. The relationship between conscientiousness trait and aggression is unclear than those having the neurotic and agreeableness trait, the reason could be that people who score low on conscientiousness are more impulsive and pay less attention to possible consequences of their actions, which makes them less deterred by the unfavorable social repercussions of aggressive and disruptive behaviors.

The current study demonstrated that the neuroticism was significantly and positively related with different dimensions of aggression. This result confirms other studies on the relationship between individual’s personality and behavioral disorder [11,48]. Generally, people with higher neurotic trait tend to be more easily agitated. These people are believed to be less emotionally stable [12,49]. Therefore, emotional instability and interpersonal conflict are more likely in those who exhibit a number of neurotic traits.

Regarding the family environment, the results indicated significant relationship of family environment with aggression, which is consistent with the results of previous studies [50,51,52]. The findings of the study revealed that family cohesion and expressiveness were negatively and significantly related to different dimensions of aggression. It could be perceived that persons with strong cohesion and expressiveness are less aggressive. Family cohesiveness and expressiveness may encourage emotion control and cognitive abilities needed to handle aggression [25]. Aggression may result from a lack of collaboration among family members, while stable family ties reduce the amount of aggression. This observation is consistent with previous research [25,53,54]. However, family conflict was found positively and significantly correlated with different dimensions of aggression. It suggests that greater the conflict in family higher the risk to engage in aggressive behavior. The finding points to the fact that people high on conflict tend to have higher tendency for aggression [15,25,55]. Specifically, better than average family environments were linked to a decrease in aggression in children; these children were functioning within the normal range and were less aggressive than their at-risk peers who lacked the advantage of highly cohesive and conflict-free homes. In other words, highly ideal family environments helped such at-risk children achieve a less aggressive developmental trajectory over time by moderating the negative effects of early aggression [25].

### Limitations and Implications of the Study

There were some limitations to the current study. First, data of the current research was obtained from AlHasa Governorate of Saudi Arabia. Data collected in this setting may thus be unique, and a replication of this study in other regions of the country may produce different outcome. Second, the convenience sampling method of sampling is not likely to be representative of all children living in other parts of the country. Therefore, further study needs representative samples in order to establish the generalizability of findings on children living in other parts of the country. Third, the present study did not find gender differences in personality traits to aggression, much research has demonstrated significant gender differences in different dimensions of aggression. For instance, whereas physical aggression is more typical in boys, relational aggression is more salient in girls [56]. Future research on gender differences across different aggression dimensions is needed to better understand the heterogeneity of aggression. Fourth, as measures used in the present study were based on the participant’s self-reports and reflections, the scores were likely to be affected by those individuals who wanted to appear more or less socially desirable. Self-report questionnaires are always susceptible to biased responses from individuals who prefer to endorse socially desirable answers. Even though attempts were made to increase objectivity by means of verbal assurances, this might not have been sufficient given that increased and decreased social desirability is strong in some individuals and because participant subjectivity is not always consciously known. Furthermore, the use of self-report measures, which are also affected by a participant’s level of self-awareness, was another limitation of this study. In future studies all sources of bias associated with self-report measures should be accounted for. Finally, measurement issues may also be one of the limitations of this study. While all of the measures demonstrated adequate to strong reliability (Coefficient Alpha), the results suggest that some of the measures warrant further examination to see whether they are applicable and appropriate to Saudi children.

Despite these limitations, this study suggested several implications for interventions to help reduce child’s aggression. First, the results of the study indicated that neuroticism trait was negatively related to children aggression. This study could help parents and teachers in reducing the negative impact of children’s high neuroticism on aggressive behavior. Parents should develop a proper concept of education and understand that family education is more important for the development of a sound personality than it is for the safety and material fulfillment of children. Parents should also focus on giving their children positive emotional responses. Positive reactions help children improve their ability to express themselves and interpersonal skills, which in turn encourages social adaptation to change their sensitivity, low self-esteem, and impulsivity and lessen the likelihood of their aggressive behavior.

Teachers should first understand the children’s personality traits and actively pay attention to highly neurotic children. Second, they can encourage children to build positive peer relationships and develop their interpersonal skills by using group counseling and club activities when delivering mental health education. Finally, educators need to pay closer attention to the behavior and mental health status of aggressive children, identify problem behaviors quickly, and implement focused interventions. To break the cycle of bullying and aggression motivated by hostility or revenge, teachers should help children in managing negative emotion and make an appropriate attribution when intervening with aggressive children [57].

Second, the family environment contributes to a child’s aggressive behavior. To foster a positive environment for a child’s behavioral development, it is crucial to offer parenting behavior interventions. An earlier empirical experiment had demonstrated that better parenting in the areas of providing more responsiveness and stimulation, reducing harsh parenting, and employing more positive discipline were effective in reducing child physical aggression in a risk sample [58]. Studies have shown that parenting behaviors, such as physical punishment and parent-child interaction, can also be influenced by a child’s behaviors [59], so it’s important to include children in the intervention and family-focused processes.

## 5. Conclusions

The present study demonstrated the differences in scores of different domains for the Big Five personality traits and family environment related to various aggression factors. Conscientiousness, agreeableness, and neuroticism factors of personality were significantly related to aggression. Family cohesion, expressiveness and conflict significantly predict the aggressive behavior among these children. A further exploration of these variables may be significant in designing interventions for children at an early stage.

## Figures and Tables

**Table 1 brainsci-12-01586-t001:** Demographic and personal characteristics of participants.

Characteristics	*N*	%
**Gender**		
Male	147	46.7
Female	168	53.3
**Age**		
<6years	41	13.0
7–9years	121	38.4
>10 years	153	48.6
**Educational level**		
Grade 1	85	27.0
Grade 2	33	10.5
**Grade 3**	33	10.5
Grade 4	32	10.2
Grade 5	46	14.6
Grade 6	86	27.3
**Family type**		
Nuclear	249	79.0
Joint	66	21.0
**Area of residence**		
Urban	295	90.5
Rural	30	9.5
**Housing status**		
Rented	98	31.1
Owned	217	68.9
**Family income**		
<5000 (Saudi Riyal)	37	11.7
5001–10,000 (Saudi Riyal)	78	24.8
10,001–15,000 (Saudi Riyal)	107	34.0
>15,001 (Saudi Riyal)	93	29.5

**Table 2 brainsci-12-01586-t002:** Means, standard deviations, and bivariate correlations of study variables.

Variables		1	2	3	4	5	6	7	8	9	10	11	12
Big Five Factors													
1	Extraversion	−											
2	Agreeableness	0.20 **	−										
3	Conscientiousness	0.11 *	0.16 **	−									
4	Neuroticism	−0.10	−0.24 **	−0.21 **	−								
5	Openness	0.13 **	0.22 **	0.15 **	−0.01	−							
Family Environment													
6	Cohesion	0.23 **	0.22 **	0.22 **	−0.18 **	0.19 **	−						
7	Expressiveness	0.22 **	0.18 **	0.07	−0.10	0.18 **	0.56 **	−					
8	Conflict	−0.16	−0.22 **	−0.25 **	0.23 **	−0.11 *	−0.44 **	−0.30 **	−				
Aggression													
9	Physical Aggression	−0.09	−0.29 **	−0.25 **	0.21 **	−0.10	−0.39 **	−0.24 **	0.46 **	−			
10	Verbal aggression	−0.07	−0.26 **	−0.20 **	0.20 **	−0.09	−0.31 **	−0.29 **	0.48 **	0.64 **	−		
11	Anger	−0.22	−0.19 **	−0.22 **	0.31 **	−0.05	−0.34 **	−0.19 **	0.45 **	0.64 **	0.65 **	−	
12	Hostility	−0.12 *	−0.13 *	−0.17 *	0.25 **	−0.09	−0.46 **	−0.41 **	0.50 **	0.50 **	0.63 **	0.61 **	
	*M*	5.57	7.60	7.50	6.20	7.50	27.95	11.27	15.80	6.62	7.46	7.20	8.03
	*SD*	1.56	1.75	2.03	2.23	1.87	5.27	2.90	5.42	3.00	2.71	3.12	2.61

* *p* < 0.05, ** *p* < 0.01.

**Table 3 brainsci-12-01586-t003:** Result of Stepwise Multiple Regression to predict Aggression from Big Five Personality Factors.

Criterion Variables	Predictors	*R* ^2^	*F* (1, 313)	*b*	*SE-b*	*β*	*t*	95% CI
Physical Aggression	AG	0.8	28.69 **	−0.49	0.09	−0.29	−5.35 **	−0.68–−0.31
	AG	0.12	22.94 **	−0.43	0.09	−0.25	−4.77 **	−0.61–−0.25
	CO			−0.31	0.07	−0.21	−3.97 **	−0.47–−0.15
	AG	0.14	16.81 **	−0.39	0.09	−0.23	−4.25 **	−0.58–−0.21
	CO			−0.28	0.08	−0.19	−3.54 **	−0.44–−0.12
	NE			0.15	0.07	0.11	2.23 *	0.00–0.29
Verbal Aggression	AG	0.06	22.59 **	−0.40	0.08	−0.25	−4.75 **	−0.57–−0.23
	AG	0.09	16.46 **	−0.36	0.08	−0.23	−4.22 **	−0.52–−0.19
	CO			−0.22	0.07	−0.17	−3.11 *	−0.37–−0.08
	AG	0.10	12.66 **	−0.32	0.08	−0.20	−3.72 **	−0.49–−0.15
	CO			−0.19	0.07	−0.14	−2.67 **	−0.34–−0.05
	NE			0.14	0.06	0.12	2.16 *	0.01–0.28
Anger	NE	0.09	33.78 **	0.43	0.07	0.31	5.81 **	0.28–0.58
	NE	0.12	21.77 **	0.38	0.07	0.27	5.08 **	0.23–0.53
	CO			−0.25	0.08	−0.16	−2.98 *	−0.41–−0.08
	NE	0.13	15.94 **	0.35	0.07	0.25	4.57 **	0.20–0.50
	CO			−0.23	0.08	−0.15	−2.75 **	−0.39–−0.06
	AG			−0.19	.09	−0.10	−1.97 *	−0.38–0.00
Hostility	NE	0.06	21.64 **	0.29	0.06	0.25	4.65 **	0.17–0.42
	NE	0.08	13.55 **	0.26	0.06	0.22	4.07 **	0.13–0.39
	CO			−0.16	0.07	−0.12	−2.27 *	−0.30–−0.02

* *p* < 0.05.** *p* < 0.01. AG = Agreeableness; CO = Conscientiousness; NE = Neuroticism.

**Table 4 brainsci-12-01586-t004:** Result of Stepwise Multiple Regression to Predict Aggression from Family Environment.

Criterion Variables	Predictors	*R^2^*	*F* (1, 313)	*b*	*SE-b*	*β*	*t*	95% CI
Physical Aggression	CO	0.21	86.52 **	0.25	0.02	0.46	9.30 **	20–0.31
	CO	0.26	55.10 **	0.19	0.03	0.36	6.60 **	0.14–−0.25
	CH			−0.13	0.03	−0.23	−4.33 **	−0.19–−0.07
Verbal Aggression	CO	0.23	96.62 **	0.24	0.02	0.48	9.83 **	0.19–0.29
	CO	0.25	54.42 **	0.21	0.02	0.43	8.53 **	0.16–0.27
	EX			−0.14	0.04	−0.15	−3.09 **	−0.23–−0.05
Anger	CO	0.20	79.54 **	0.26	0.02	0.45	8.91 **	0.20–0.31
	CO	0.22	45.66 **	0.21	0.03	0.37	6.69 **	0.15–27
	CH			−0.10	0.03	−0.17	−3.09 **	−0.16–−0.03
Hostility	CO	0.25	105.07 **	0.24	0.02	0.50	10.25 **	0.19–28
	CO	0.32	76.07 **	0.19	0.02	0.41	8.45 **	0.15–24
	EX			−0.26	0.04	−0.29	−5.95 **	−0.34–−0.17
	CO	0.34	54.91 **	0.17	0.02	0.35	6.99 **	0.12–0.22
	EX			−0.18	0.05	−0.19	−3.48 **	−0.28–−0.07
	CH			−0.09	0.03	−0.18	−2.96 **	−0.14–−0.30

* *p* < 0.05.** *p* < 0.01. CO = Conflict; CH = Cohesion; EX = Expressiveness.

## Data Availability

The data that support our findings can be found through directly asking the corresponding author.

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
