# Peer review of "Personality Traits and Family Environment: Antecedents of Child Aggression"

_brainsci, 2022, doi:10.3390/brainsci12111586_

Round 1
Reviewer 1 Report
Dear author and editor
I consider this a relevant, well-planned and well-developed research. It is simple and clear. True, the sample is not very large, but it is statistically significant and perfectly valid. I only miss a clearer and more detailed description of the sampling process, as well as how the actions were carried out to maintain ethics. It is clear to me that it went through a committee since it appears at the end of the research, but it is missing in the study.
Also, the hypotheses could be rewritten in a clearer way, as it is a little difficult to understand them. Finally, the literature review could benefit from citing longitudinal studies and meta-analyses, as they will give it a point of superior richness.
Finally, I would like to congratulate the research team. I know that it is not easy to publish studies with small samples (although representative and statistically valid), but I consider the quality of the data and their veracity to be more important criteria.
Reviewer 2 Report
1) I commend the authors for giving some indication of how aggression (and the various forms of aggression) seem to evolve over the course of development. For example, we know that physical aggression tends to peak in early adolescence, while verbal and social forms of aggression become more salient in adolescence.
2) What theoretical basis can explain the relationship between family context and aggressive behavior in children?
3) The methodology should be better explained. In particular, the way the study was conducted (procedures) should be examined. I have some reservations about young children being able to independently complete self-report questionnaires. Also, I would like to understand how you ensured the privacy of the participants and whether the research project was approved by the ethics committee.
4) In my opinion, the discussion lacks theoretical insight. I would ask the authors to try to explain the relationship between the variables in light of the theoretical framework they follow.
5) The paper should be revised, paying attention to the terms that indicate a causal relationship between the variables. Your study is in fact a cross-sectional study.
6) The section on the limitations of the research needs to be updated and expanded upon. You have written little on this point.
7) Practical implications?
Reviewer 3 Report
Thank you for the opportunity to review the manuscript, but I have some comments and suggestions. The topic is interesting, but the manuscript is not fluid for good understanding:
In the abstract there are 316 participants but, in the method, there are 315.
Better highlight research hypotheses, reporting them later in discussions
On the materials highlight only the scales used and not the various existing versions.
I would include descriptive analysis of the overall sample in the results.
For better understanding, I recommend simplifying the inserted tables by removing from the text or table what has already been highlighted.
Why was no mediation or moderation model proposed to better understand the direction of the results obtained?
Resume discussions considering the hypotheses proposed by the study.
In the conclusions it is useful to add the applications of this study and what it could be used for.
Best regards.
Round 2
Reviewer 2 Report
I accept the manuscript in current form! Thank for your revision!
Author Response
Your comments have made our manuscript better in terms of quality. We appreciate your comments and valuable advice. Thank you
Reviewer 3 Report
Ok.
For future work, verify that you do not overlap the data in this manuscript with those with mediation models.
Author Response
Comment: For future work, verify that you do not overlap the data in this manuscript with those with mediation models.
Response: Thanks for this insightful advice, surely we will not include the results of the current study in our future work.
we thank you for your valuable comments, these comments have increased the quality of our manuscript.